# Effectiveness of a Participatory Program for Improving the Cardiovascular and Cerebrovascular Health of Older Farmers in Rural Korea

**DOI:** 10.3390/ijerph20043210

**Published:** 2023-02-12

**Authors:** Ki-Youn Kim, Juhye Jin, Yeon-Ha Kim

**Affiliations:** 1Department of Safety Engineering, Seoul National University of Science and Technology, Seoul 01811, Republic of Korea; 2Department of Nursing, Korea National University of Transportation, Jeungpyeong 27909, Republic of Korea

**Keywords:** community-based participatory research, aged, farmers, coronary artery disease, cerebrovascular disorder

## Abstract

Background: The aim of this study was to evaluate the effectiveness of a participatory approach to the cardiovascular and cerebrovascular (CCV) health of older farmers in rural Korea. Methods: A nonequivalent control group pretest–posttest design was used. Participants included 58 farmers aged ≥ 60 years who were assigned to either an experimental group (n = 28) or a comparative group (n = 30). The experimental group took part in the participatory program for CCV health, while the comparative group received a conventional lecture program for CCV health. The two groups, from pretest to posttest, were compared using the generalized estimating equation (GEE) method. Results: The participatory program showed a greater effect over time than the conventional lecture program for health empowerment (Χ^2^ = 7.92, *p* = 0.005) and self-efficacy in managing CCV health (Χ^2^ = 5.94, *p* = 0.015). The average rate of implemented improvements after 3 months was 88.9%, showing that the participatory program was successful. Conclusions: The participatory program for CCV health was an effective intervention for the empowerment and self-efficacy of older farmers in managing their own CCV health. Therefore, we recommend replacing lectures with participatory methods in CCV health programs for older farmers.

## 1. Introduction

In Korea, the proportion of agricultural workers aged 65 years or older has risen significantly, from 40.3% in 2016 to 46.8% in 2021 [1], due to an increase in the total population aged 65 and older. South Korea is predicted to become a super-aged society by 2025 [2]. Since older farmers work long hours in hazardous, stressful, and physically demanding work environments [3], their health concerns warrant recognition and attention. However, there has been limited research on the health and safety of older farmers focusing on their cardiovascular and cerebrovascular (CCV) health.

CCV diseases are the most common diseases among elderly adults and have been a leading cause of death in South Korea for almost a century [4]. Rural elderly adults have a higher rate of CCV diseases [4] than the urban population due to the high incidence of lifestyle diseases such as hypertension, diabetes, and hyperlipidemia [5]. Recent studies have demonstrated that changes in lifestyle can prevent CCV diseases through a multifactorial approach including control of hypertension, diabetes, hyperlipidemia, tobacco use, alcohol intake, physical activity, healthy diet, mental health burdens, stress, and poor sleep, as well as pharmacological strategies [5,6,7,8,9,10,11]. However, most programs to improve CCV health have focused on single behavioral risk factors (e.g., diet, smoking, physical activity) or single clinical risk factors (e.g., hypertension, diabetes, hypercholesterolemia) and are presented through conventional lecture formats [12,13].

It is problematic that CCV health programs have predominantly used conventional lecture interventions because they are short-term interventions and insufficient to provide a sustained effect on lifestyle changes [6]. New methods are needed to improve the ability of older farmers to identify CCV health risks and empower them with self-determined and sustainable health strategies to improve their CCV health behaviors [14]. This strategy must also instill confidence in their capacity to positively control their CCV health (i.e., self-efficacy) [15].

When implementing agricultural workplace health interventions, success is more likely if a participatory method is utilized because of its learning-by-doing approach, exchange of positive experiences, non-expert driven interventions, and action plans that build upon local practice [16,17]. Effective participatory interventions require a structured, organized approach involving all levels of an organization (i.e., individual, group, leader, and organizational), and optimal use of on-site resources [14]. Thus, we developed a participatory program for CCV health (PAP-CCVH) that involved all organizational levels and we aimed to determine whether the older farmers felt empowered to improve their CCV health behaviors, achieve self-efficacy, and sustain their improvements. Most CCV health programs measure clinical outcomes, whereas few programs assess behavioral or psychological changes, such as health empowerment and self-efficacy [18], that are needed to implement the actions that ultimately manage CCV health. The reason for this may be that uncertainty regarding which health problems to target and which actions to improve must be addressed first [19].

Therefore, the aim of this study was to evaluate the effectiveness of a PAP-CCVH compared to the conventional lecture program for CCV health (CLP-CCVH) among farmers ≥ 60 years old in rural Korea. The hypothesis of this study was that older farmers who received the PAP-CCVH, when compared to farmers who received the CLP-CVVH, would show (1) higher health empowerment, (2) higher self-efficacy in managing CCV health, and (3) improved CCV health outcomes after making lifestyle behavior changes.

## 2. Materials and Methods

### 2.1. Participants and Procedures

This study had a nonequivalent control group pretest–posttest design in which the controls (comparative group) and an experimental group received different interventions.

The participants were older farmers in E County, Chungbuk Province, South Korea who farmed various agricultural products including rice, wheat, vegetables, and fruit trees. They were self-referred or referred by their community health center.

Farmers ≥ 60 years old with normal cognitive function (Mini Mental State Examination-Korea [MMSE-K] test score ≥ 24), who were ambulatory and able to communicate and read were included. The exclusion criteria were cognitive impairment (MMSE-K score < 24), treatment with antidepressant or antipsychotic medications, neurological diseases such as cerebral infarction or Parkinson’s disease, addiction to drugs or alcohol, and cardiovascular and cerebrovascular disorders.

The sample size was calculated using the G*Power 3.1 program (Düsseldorf, Germany) with repeated-measure analysis of variance. The level of significance (α) was set at 0.05, with a statistical power (1 − β) of 0.95, and intra-class correlation of 0.05. The number of groups was 2, the number of measurements was 2, and the effect was 0.44. The effect size chosen for sample size calculation was based on a similar intervention health program among older adults [20]. The minimum sample size was 27 for both the experimental group and the comparative group. Intervention studies of elderly adults often have a high dropout rate. Therefore, we aimed for 33 older farmers in each group. The final sample size was 28 in the experimental group and 30 in the comparative group (total 58 participants). The main reason for dropout was due to incomplete questionnaires.

The PAP-CCVH was provided to the experimental group and the CLP-CCVH was provided to the comparative group. The participants did not know which interventions have been assigned (Figure 1).

### 2.2. Development of a Participatory Program for Cardiovascular and Cerebrovascular Health (PAP-CCVH)

A specifically tailored PAP-CCVH for older farmers was designed based on the principles of participatory action-oriented training programs, which include building upon local practice, focusing on achievements, using a learning-by-doing approach, encouraging exchange of experiences, and promoting involvement. The principles of this methodology include using (1) a structured approach with participatory training tools such as good examples and action checklists, (2) a trained facilitator to encourage participants to step up their improvement actions, and (3) work groups to generate action plans [17].

#### 2.2.1. Develop an Action Checklist (Organizational Level)

A CCV health action checklist tailored to older farmers was developed (Table 1). This learning-by-doing action checklist was developed to help farmers distinguish between the positive behaviors and the “to be improved” behaviors related to their CCV health. It was developed by modifying the action checklist for CCV disease developed by Yoon et al. [21]. The authors added two oral hygiene checklist items to the 62 original items due to multiple reports that poor oral health is associated with cardiovascular disease in older populations [22]. Next, 27 primary healthcare service workers took a one-day facilitator training workshop and selected the priority items that E County older farmers should perform successfully. They reached a consensus on 31 items. Finally, content validity of the 31 items was verified by two medical college professors and two community nursing professors who were experts in developing participatory-action checklists. The 31 items were categorized into nine main areas.

#### 2.2.2. Train Facilitators (Organizational and Leader Level)

Primary healthcare service providers from each town in E county were trained to facilitate and support the participants. They were trained to enable participants to (1) use the action checklist, (2) exchange positive experiences related to their own CCV disease prevention actions as part of group work, (3) develop action plans based on their own capabilities and functional health, and (4) implement their plans as actions. A facilitator should not impose their own subjective ideas on the participants; instead, a facilitator answer questions and motivate the participants. After the participatory group work, the facilitators provided health coaching and monitored whether the participants were implementing their action plans.

#### 2.2.3. Conduct Group Work (Group, Individual Level)

The main program was conducted as a group over a half-day course (180 min) (Table 2). Participants were divided into small teams of 3~6 members—participants and their affiliated primary health care service provider (facilitator). The program progressed with (1) an opening presentation of symptoms, emergency actions, and treatments for CCV disease; (2) learning to measure blood pressure, blood sugar, and lipid levels; (3) reviewing the action checklist as a self-help guide; (4) learning to identify positive CCV health behaviors versus risky behaviors that need to be improved; and, (5) group discussions that enabled group members to share their experiences with CCV health management.

Participants individually identified their positive CCV health behaviors and their risky behaviors that needed to be improved. The behaviors that needed to be improved within 3 months were incorporated into action plans based on their individual capabilities and functional health, starting with small steps.

The implemented actions were evaluated at 3 months. During those 3 months, the participants received on-site health coaching from their facilitator who also monitored whether they were implementing their action plans.

### 2.3. Development of a Conventional Lecture Program for Cardiovascular and Cerebrovascular Health (CLP-CCVH)

The CLP-CCVH for older farmers was designed as an expert-driven knowledge delivery method, which included the same program content as the experimental group: (1) an opening presentation on the symptoms, emergency actions, and treatment of CCV disease, (2) learning to measure blood pressure, blood sugar, and lipid levels, and (3) lifestyle behavior management. The CLP-CCVH was conducted over 3 weeks, with 60 min presentations once a week (180 min total). The education time per week was based on previous research [20] (Table 2).

### 2.4. Measurements

The data were collected via a questionnaire survey and the results of action plan implementations. A self-administered questionnaire was designed to obtain the participants’ personal characteristics, as well as their degree of health empowerment and self-efficacy in managing CCV health. The Korean version of the health empowerment scale (K-HES) is a validated 8-item questionnaire with questions that ask about an individual’s degree of autonomy and self-determination in managing their general health [23]. The K-HES was modified from the Diabetes Empowerment Scale-Short Form [24]. The K-HES items were scored on a 5-point Likert scale ranging from “strongly disagree” (1) to “strongly agree” (5). A higher value indicated a stronger degree of empowerment. This instrument demonstrated good internal consistency with a Cronbach’s α of 0.867.

A self-efficacy scale for managing CCV health was developed and modified by Park and Kim [25] into 15 items, each starting with “How confident are you that you can...?” The questions addressed psychological adaptation (3 items), health management (2 items), eating habits (2 items), regular exercise (2 items), smoking (2 items), alcohol intake (2 items), stress (1 item), and sleep (1 item). This scale was based on the General Self-Efficacy Scale [26] and a cardiovascular disease risk perception scale [27]. It was scored on a 5-point Likert scale ranging from “strongly disagree” (1) to “strongly agree” (5). A higher value indicated a strong self-efficacy in managing CCV health. This instrument demonstrated good internal consistency with a Cronbach’s α of 0.799.

Three months after the intervention, the implementation rate of the action plans was surveyed in the PAP-CCVH group only.

### 2.5. Data Analysis

All statistical analyses were performed using SPSS version 23.0 (IBM Corp., Armonk, NY, USA). Means (standard deviation) and frequencies (%) were used to analyze descriptive data. The homogeneity of general characteristics and outcome variables was analyzed using the chi-square test, Fisher exact test, and *t*-test. The outcome variables were health empowerment and self-efficacy in managing CCV health. The normality of the outcome variables was tested using the Shapiro–Wilk test and the normality was not satisfied. Thus, the differences between pretest and posttest were analyzed using the Wilcoxon signed rank test and a comparison of the two groups from pretest to posttest was evaluated using the generalized estimating equation (GEE) method.

### 2.6. Data Collection

This study was approved by the Institutional Review Board (IRB) of the researchers’ affiliated university (Korea National University of Transportation, IRB 2019-14). All participants were assured of confidentiality and anonymity before providing written consent. All were informed of the purpose of the study and their right to withdraw without penalty.

In both the experimental group and the comparative group, data were collected just before and after the intervention was conducted. Implemented actions were evaluated only in the experimental group 3 months after the intervention. It was conducted from May 2019 to October 2019.

## 3. Results

### 3.1. Homogeneity Test of General Characteristics and Outcome Variables

The general and outcome variables of the study population in the pretest are shown in Table 3. Differences in gender, age, education, marital status, perceived health status, smoking status, and alcohol consumption were not significant between the experimental group and the comparative group. Health empowerment and self-efficacy in managing CCV health showed no significant differences between the experimental group and the comparative group. This indicated homogeneity between the two groups.

### 3.2. Comparison of Health Empowerment among Groups

A significant increase in health empowerment levels was seen from the pretest to the posttest in both the experimental group (Z = −3.51, *p* < 0.0001) and the comparative group (Z = −2.28, *p* = 0.023) (Table 4). The findings for the effects of group, time, and the interaction between group and time are presented in Table 4. Significant differences were not found according to group (*p* = 0.094) but were observed according to time (*p* < 0.001) and the interaction between group and time (*p* = 0.005).

### 3.3. Comparison of Self-Efficacy in Managing CCV Health among Groups

Self-efficacy in managing CCV health also showed significant improvements from the pretest to posttest in both the experimental group (Z = −3.75, *p* < 0.001) and the comparative group (Z = −3.41, *p* = 0.001) (Table 4). The findings for the effects of group, time, and the interaction of group and time are shown in Table 4. Significant effects were found according to group (*p* = 0.001), time (*p* < 0.001), and the interaction effect of group and time (*p* = 0.015)

### 3.4. Proposed and Implemented Action Plans

The proposed and implemented action plans are presented in Table 5. The total number of action plans in every action checklist area was 352. The average rate of successfully implementing improvements at three-month evaluation was 88.9% (min. 84.1%, max. 100.0%).

## 4. Discussion

This paper presents the design and protocol of a nonequivalent control group pretest–posttest to evaluate the effectiveness of a participatory program versus a conventional lecture program for promoting the CCV health of older farmers.

The levels of health empowerment showed significant improvement after intervention compared to the baseline levels in both groups (PAP-CCVH and CLP-CCVH). Although both intervention methods showed improvement in health empowerment, the participatory method showed greater effect over time than the conventional lecture method. A previous study of cardiovascular health programs for adults reported that when good observational and experiential learning methods, action plans, and counseling were utilized to empower participants to reach their personal health goals, cardiovascular disease risk factors were significantly improved [28]. Baydur et al. reported that a participatory training method empowered and improved the capacity of participants to change lifestyle behaviors of farmers [16]. However, to the best of our knowledge, no previous studies have estimated the effectiveness of health empowerment for the CCV health of older adults [29]. Our study results indicated that participatory methods were more effective than conventional lecture methods for empowering older farmers to improve their CCV health.

The levels of self-efficacy in managing CCV showed significant improvement after intervention compared to the baseline levels in both groups (PAP-CCVH and CLP-CCVH). Although both intervention methods showed improved self-efficacy in managing CCV health, the participatory method showed a greater effect over time compared to the conventional lecture method. This result was consistent with a previous study of hypertension self-management programs in rural older adults that used similar intervention methods, such as group discussions to set goals and solve problems [30]. That study showed a significant 22.2% increase in self-efficacy compared with other studies that used expert-led hypertension interventions and only showed a 10% increase [31]. Based on these previous research results, we assumed that the group work conducted in the PAP-CCVH contributed to the higher increase in self-efficacy when compared to the CLP-CCVH. Group discussion allows participants to exchange their positive experiences and figure out how to create their own plan of action [15]. Action checklists also increase participants’ confidence that they can change their lifestyle behaviors [32]. Previous studies reported that perceived self-efficacy in cardiovascular disease was positively associated with physical exercise, food choices, smoking habits, and health-related quality of life [30]. The action checklist used in our study listed useful and positive lifestyle behaviors and helped participants identify their own positive CCV health behaviors and those that needed improvement so that they could succeed in executing their action plans. Thus, when providing CCV health interventions to older farmers, a tailored action checklist needs to replace printed references as educational material. The action checklist must present available improvement options and serve as an aid to decision-making when participants are selecting important improvement actions [17].

In our study, 88.9% of proposed action plans were implemented within 3 months after the PAP-CCVH. A previous participatory study of traditional retail market workers [32] that used a similar action checklist reported an 82.7% implementation rate. The higher rate in our study may have been because the learning-by-doing action checklists used in our study were shortened to fit the needs and lifestyle of older people and so that the goals would be acceptable to the participants. Action checklists allow participants to take control of the determinants of their own health and produce practicable changes [17,32]. Another reason for the high rate of success in our study was that the PAP-CCVH combined with the support of a facilitator improved the farmers’ health empowerment and self-efficacy in managing CCV health and maintained the effect of the intervention for a prolonged period of time. The primary health care service providers/facilitators were existing organizational resources who worked closely with the older farmers in the PAP-CCVH, counseling and monitoring them for 3 months while they implemented their action plans. One study reported that expert-led educational methods were not effective and required follow-up sessions to sustain the effect of the intervention [33]. In our study, the role of the facilitator was to allow participants to determine their own health behavior improvement goals and, during follow-up visits, to encourage the older farmers to implement their action plans. This result highlights the importance of using existing organizational and leadership resources. Therefore, the older farmers who participated in this PAP-CCVH could play important roles as future facilitators and peer leaders in their communities. It will also be necessary to train more farmers as facilitators for the PAP-CCVH to expand throughout other agricultural areas.

## 5. Conclusions

The PAP-CCVH for older farmers, which involved developing an action checklist, facilitator training, group work, and building action plans and implementing them, resulted in significant improvements in health empowerment and self-efficacy in managing CCV health compared with the CLP-CCVH. The PAP-CCVH needs to replace the lecture method of providing CCV health programs. To enhance health care in agricultural settings, more farmers must be trained to be facilitators to expand the PAP-CCVH throughout other agricultural areas, and the tailored action checklist needs to replace printed reference materials as an educational resource.

The limitations of this study are as follows. First, the study subjects were from a geographically limited area; therefore, the findings of this study may not be generalizable to other Korean farmers. Second, the farmers who committed to participating in this study were sufficiently interested and motivated to change their health behaviors. Therefore, self-selection bias may have been present. Third, although the aim of this study was to investigate proactive behavioral management to prevent CCV disease in older farmers, clinical measurements are needed in future studies.

## Figures and Tables

**Figure 1 ijerph-20-03210-f001:**
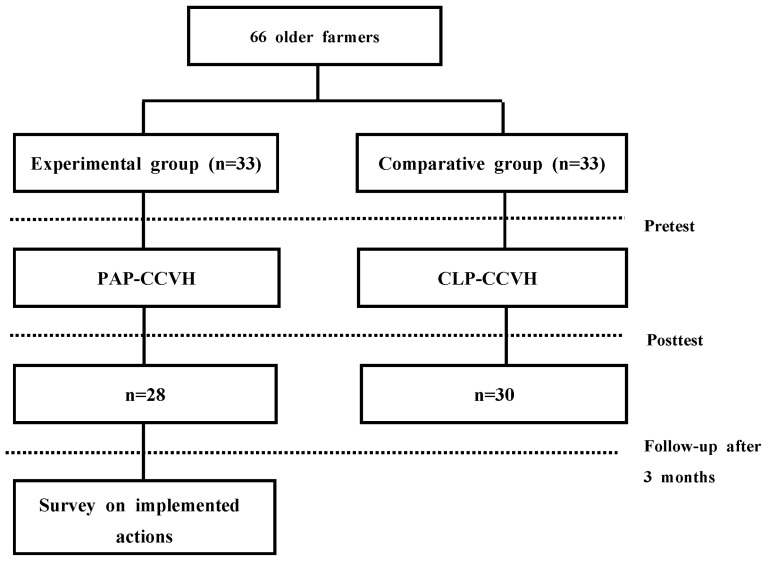
PAP-CCVH, participatory program for cardiovascular and cerebrovascular health (experimental group); CLP-CCVH, conventional lecture program for cardiovascular and cerebrovascular health (comparative group). Flow chart of participants.

**Table 1 ijerph-20-03210-t001:** Action checklist for management of the cardiovascular and cerebrovascular health of older farmers.

	Checklist Items
	Hypertension management
1.	Take antihypertensive medication as prescribed by a doctor.
2.	Measure blood pressure at least 3 times a week and record the results (stay below 130/80 mmHg).
3.	Receive regular urine tests, kidney function tests, and electrocardiograms (ECGs).
	Diabetes management
4.	Take diabetes medications as prescribed by a doctor.
5.	Know the normal blood sugar levels, such as fasting blood sugar (70–130 mg/dL), 2-h postprandial blood sugar (90–180 mg/dL), hemoglobin A1C (≤6.5%), and strictly manage your blood sugar with the goal of maintaining normal levels.
6.	Receive regular screening for diabetes complications: (HbA1C test every 3 months), fundus (retina) test every year, kidney function test (microalbuminuria test) every year, peripheral sensory test, and carotid artery ultrasonogram.
	Dyslipidemia management
7.	Take cholesterol-lowering medications as prescribed by a doctor.
8.	Receive regular hyperlipidemia (dyslipidemia) tests.
9.	Know the normal lipid levels, such as total cholesterol (<200 mg/dL), triglyceride (<150 mg/dL), LDL cholesterol (<100 mg/dL), and HDL cholesterol (≥60 mg/dL), and manage the levels.
	Diet management
10.	Do not eat soups made with salty foods, such as soybean paste stew and kimchi stew. Eat bland foods as much as possible.
11.	Drink more than 1.5 L of water per day.
12.	Regularly eat food with a low glycemic index: multigrain rice, beans, vegetables, seaweed, and raw fruits with low sugar content.
13.	Try to avoid foods high in saturated or trans fats, such as animal fats, fried foods, and fried pancakes.
14.	To build stronger muscles, exercise (for example, walking up the stairs and doing sit-ups) and eat protein-rich foods: chicken breasts, egg whites, fish, and lean cuts of beef or pork.
15.	Eat a variety of foods in even proportions, such as rice, other grains, vegetables, fruits, milk, dairy products, meat, fish, eggs, and beans (all rich in iron, zinc, calcium, and vitamins).
16.	Chew slowly many times when eating.
	Exercise management
17.	Avoid working in the same posture for a long time. Use a timer to stretch your body every hour, consciously stretching the chest and back muscles to improve blood circulation.
18.	Do aerobic exercise regularly, at least 3 times a week for at least 30 min (a total of 150 min or more per week) and move around as much as possible. Walking dilates blood vessels, which improves blood circulation, enhances immunity, promotes removal of accumulated substances from the body, and suppresses aging.
19.	Do strength training (free gymnastics, yoga, dumbbells, sit-ups) consistently for several minutes.
20.	Maintain a correct posture while sitting or walking, and consciously stretch your chest and back muscles. A correct posture improves blood circulation by 25%, increases calorie consumption, and improves work efficiency.
	Stress and sleep management
21.	Have conversations with others, empathize with them, and praise others as you listen to their stories. Talking openly is the best way to relieve built-up stress.
22.	Do abdominal breathing at least 3 times a day.
23.	Laugh for more than 10 s, 3 times a day. Laughing loudly through exhalation releases good hormones, removes waste products from the body, relaxes blood vessels, and helps improve immunity.
24.	Regulate overall health with a regular soaking bath. It has been reported that the average life expectancy of those who bathe regularly was 20 years longer than that of those without a bath/sauna culture.
25.	Try to sleep at least 6–8 h.
	Oral management
26.	Brush teeth immediately after eating.
27.	For the purposes of prevention and treatment, visit the dentist regularly for care.
	Drinking habit management
28.	Do not drink alcohol every day. If drinking, then drink below your personal limits. Excessive drinking increases fatigue, so take breaks of 2–3 days with no alcohol. Alcohol and its byproduct acetaldehyde are class 1 carcinogens. Therefore, excessive drinking increases the risk of cancer.
29.	Drink slowly and drink water from time to time. Alcohol is detoxified and excreted in water.
	Smoking habit management
30.	Let people around you know if you are trying to quit smoking.
31.	Exercise (strolling, walking) or engage in hobbies to reduce the desire to smoke.

**Table 2 ijerph-20-03210-t002:** Participatory and expert-driven programs for cardiovascular and cerebrovascular (CCV) health.

Session	Theme	Program Content	Pedagogical Strategy	Time (minutes)
Participatory program for cardiovascular and cerebrovascular health (PAP-CCVH)				
1.	Opening presentations	∙Symptoms, emergency actions, and treatment of CCV disease	Lecture	30
	Self-management	∙Learn to measure blood pressure, blood sugar, and lipid levels	Facilitator support	30
2.	Learning by doing	∙Practice action checklist ∙Identify positive CCV health behaviors and risky behaviors that need to be improved	Training tool Facilitator support	60
3.	Encourage exchange of experiences	∙Exchange good experiences related to implementing the CCV disease prevention actions	Group discussion Facilitator support	30
	Promote individual involvement	∙Plan improvement actions	Individual mapping Facilitator support	30
Conventional lecture program for cardiovascular and cerebrovascular health (CLP-CCVH)				
1.	Opening presentations Self-management	∙Symptoms, emergency actions, and treatment of CCV disease ∙Learn to measure blood pressure, blood sugar, and lipid levels	Expert lecture	60
2.	CCV health management Self-management	∙Management of diet, exercise, stress, sleep, oral hygiene, alcohol intake, and smoking ∙Learn to measure blood pressure, blood sugar, and lipid levels	Expert lecture	60
3.	CCV health management	∙Management of diet, exercise, stress, sleep, oral hygiene, alcohol intake, and smoking	Expert lecture	60

**Table 3 ijerph-20-03210-t003:** The homogeneity of general characteristics and outcome variables.

Characteristics	Categories	n (%), Mean ± SD		Χ^2^/t	*p*
		Exp. (n = 28)	Comp. (n = 30)		
Gender	Male	4 (14.3)	4 (13.3)	0.11	0.916
	Female	24 (85.7)	26 (86.7)		
Age (years)	60–69	13 (48.1)	10 (33.3)	1.29	0.255
	70–80	14 (51.9)	20 (66.7)		
Education	≤Elementary school	7 (22.3)	10 (33.3)	2.20	0.332
	Middle school	10 (37.0)	6 (20.0)		
	≥High school	11 (40.7)	14 (46.7)		
Spouse	Yes	19 (67.9)	22 (73.3)	0.21	0.647
	No	9 (32.1)	8 (26.7)		
Perceived health status	Excellent	1 (3.6)	0 (0)	1.48 ^+^	0.891
	Good	4 (14.3)	4 (13.3)		
	Moderate	16 (57.1)	20 (66.7)		
	Bad	7 (25.0)	6 (20.0)		
	Very bad	0 (0)	0 (0)		
Smoking	Yes	0 (0)	2 (6.7)	1.93	0.164
	No	28 (100)	28 (93.3)		
Alcohol	Yes	8 (28.6)	14 (46.7)	2.01	0.156
	No	20 (71.4)	16 (53.3)		
Health empowerment		3.49 ± 0.69	3.42 ± 0.45	5.20	0.653
Self-efficacy in managing CCV health		3.90 ± 0.43	3.68 ± 0.42	0.05	0.052

^+^ Fisher’s exact test; Χ^2^, chi-square; t, t-value; Exp., experimental group; Comp., comparative group.

**Table 4 ijerph-20-03210-t004:** Comparison of variables among groups using a generalized linear model.

Characteristics	Categories	M ± SE		Z(*p*) ^+^	Sources	Χ^2^*(p)* ^++^
		Pretest	Posttest			
Health empowerment	Exp.	3.49 ± 0.11	3.99 ± 0.12	−3.51 (<0.001)	Group	2.80 (0.094)
	Comp.	3.42 ± 0.10	3.57 ± 0.11	−2.28 (0.023)	Time	28.70 (<0.001)
					G × T	7.92 (0.005)
Self-efficacy in	Exp.	3.90 ± 0.08	4.30 ± 0.08	−3.75 (<0.001)	Group	10.37 (0.001)
managing CCV health	Comp.	3.68 ± 0.07	3.85 ± 0.08	−3.41 (0.001)	Time	37.53 (<0.001)
					G × T	5.94 (0.015)

^+^ Wilcoxon signed rank test; ^++^ generalized estimating equation; M, mean; SE, standard error; CCV, cardiovascular and cerebrovascular; Exp., experimental group; Comp., comparative group; G × T, group × time.

**Table 5 ijerph-20-03210-t005:** Proposed and implemented action plans.

Type of Action Plan	Proposed Action Plans (n)	Implemented Action Plans (n)	Rate of Implemented Improvements (%)
Hypertension management	47	41	89.1
Diabetes management	37	31	86.2
Dyslipidemia management	31	26	84.1
Diet management	78	71	92.2
Exercise management	47	41	89.1
Stress and sleep management	80	73	92.2
Oral hygiene management	30	30	100.0
Drinking habit management	2	0	0
Smoking habit management	0	0	0
Total	352	313	88.9

## Data Availability

Not applicable.

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
