# Peer review of "Effectiveness of a Participatory Program for Improving the Cardiovascular and Cerebrovascular Health of Older Farmers in Rural Korea"

_ijerph, 2023, doi:10.3390/ijerph20043210_

Round 1

Reviewer 1 Report

This is a very interesting study. The paper is well written. I have some questions for the authors.

1.       Study participants were divided into two groups but was not explained how actually it was performed. It should be added in the method section.

2.       Figure 1 is very informative and useful. However, it would be good to know how 5 participants from the experimental and 3 participants from comparative study dropped out. It is especially interesting because follow-up after 3 months was performed only in the experimental group. For more clearly presentation of figure 1 it should be noted in the explanation below that PAP-CCVH, participatory program is experimental, and CLP-CCVH, conventional lecture program is comparative group.

3.       Also, it would be useful to know did the participants and researcher know to which of the investigated groups belonged, and how the data analysis was performed, did the statistician know to which of the groups' participants belonged to.

4.       In the table 3 abbreviation Χ2 /t should be explained.

Author Response

Dear reviewers and International Journal of Environmental Research and Public Health

I would like to express our sincere gratitude for your thorough consideration and scrutiny over our manuscript. Through the accurate and keen comments made by the reviewers, the critical points at issue in the overall manuscript were discovered and subsequently corrected. After receiving the reviewers’ criticisms, my colleagues and I have revised the manuscript in order to achieve the proper scientific and literary levels required by the reviewers and International Journal of Environmental Research and Public Health. I hope this revised manuscript will be considered positively and be accepted by International Journal of Environmental Research and Public Health. Our responses to the reviewer’s comments are as follows:

Response to Reviewer Comments

Point 1:  Study participants were divided into two groups but was not explained how actually it was performed. It should be added in the method section.

Response: Thank you for your suggestion. We have added ‘The PAP-CCVH was provided to the experimental group and the CLP-CCVH was provided to the comparative group. The patients did not know in advance the intervention assignment. A block design was utilized then randomized into either experimental or comparative group’ in page 3 L11.

Point 2: Figure 1 is very informative and useful. However, it would be good to know how 5 participants from the experimental and 3 participants from comparative study dropped out.

Response: Thank you for your suggestion. We have added ‘The main reason for dropout was due to incomplete questionnaires.’ In page 3 L10

Point 3: figure 1 should be noted in the explanation below that PAP-CCVH, participatory program is experimental, and CLP-CCVH, conventional lecture program is comparative group.

Response: Thank you for your suggestion. We have added the explanation in figure 1. 

Point 4:   it would be useful to know did the participants and researcher know to which of the investigated groups belonged, and how the data analysis was performed, did the statistician know to which of the groups' participants belonged to.

Response: Thank you very much for your suggestion. We have added ‘The participants did not know which interventions have been assigned’ in page 3, L12.

Point 4: In the table 3 abbreviation Χ2 /t should be explained.

Response: Thank you for your suggestion. We have explained the abbreviation Χ2 /t in table 3.

Thank you very much for your review.

Reviewer 2 Report

Hello,

There are several studies that show us that educational interventions contribute to the improvement of patients' quality of life. Therefore, your study is relevant when you propose to evaluate the effectiveness of a direct educational intervention in elderly farmers for cardiovascular and cerebrovascular health and compare the results of this direct intervention with the results of another non-direct intervention in lecture format in a group with the same characteristics. Regarding the two groups (experimental and comparative), I have doubts about the inclusion criteria of the participants! Were the two groups older people with and without disease? Or were farmers randomly invited to participate in the study, regardless of their CCV health status?

In the Materials and Methods chapter, I suggest first stating what kind of study was done and what was the target population.  And then the inclusion and exclusion criteria of the participants.

Point 2.5 should come at the end of the Materials and Methods chapter and should be reworded to exclude the information that is also at the end of the article.

The results of the study are well presented and clearly show that there are different outcomes when different interventions are done in the two study groups.

As for the discussion, it would have been pertinent to compare the results with more recent studies to make their conclusions more robust.  More than half of the references used are over 10 years old. Perhaps it would have been pertinent to do more research on studies similar to yours and improve the framing of the topic and discussion.

Good job!

Author Response

Dear reviewers and International Journal of Environmental Research and Public Health

I would like to express our sincere gratitude for your thorough consideration and scrutiny over our manuscript. Through the accurate and keen comments made by the reviewers, the critical points at issue in the overall manuscript were discovered and subsequently corrected. After receiving the reviewers’ criticisms, my colleagues and I have revised the manuscript in order to achieve the proper scientific and literary levels required by the reviewers and International Journal of Environmental Research and Public Health. I hope this revised manuscript will be considered positively and be accepted by International Journal of Environmental Research and Public Health. Our responses to the reviewer’s comments are as follows:

Response to Reviewer Comments

 Point 1: were farmers randomly invited to participate in the study, regardless of their CCV health status?

Response: Thank you for your answer. We have revised the exclusion criteria sentence. cardiac disorder into “cardiovascular and cerebrovascular disorders.” In Page 2

Point 2: Materials and Methods chapter, first stating what kind of study was done and what was the target population. And then the inclusion and exclusion criteria of the participants.

Response: Thank you for your suggestion. We have reorganized the sentence.

  1. “This study had a nonequivalent control group pretest-posttest design in which the controls and an experimental group received different interventions.”
  2. “The participants were older farmers in E County, Chungbuk Province, South Korea who farmed various agricultural products including rice, wheat, vegetables, and fruit trees. They were self-referred or referred by their community health center...”
  3. Farmers ≥ 60 years old with normal cognitive function (Mini Mental State Examina-tion-Korea [MMSE-K] test score ≥24), who were ambulatory and able to communicate and read were included.
  4. The exclusion criteria were cognitive impairment (MMSE-K score < 24), treatment with an-tidepressant or antipsychotic medications, neurological diseases such as cerebral infarc-tion or Parkinson disease, addiction to drugs or alcohol, and cardiovascular and cerebro-vascular disorders.

 Point 3: 2.5 should come at the end of the Materials and Methods chapter and should be reworded to exclude the information that is also at the end of the article.

Response: Thank you for your suggestion. We have reorganized 2.5 Data collection at the end of the Materials and Methods.

Point 4 Perhaps it would have been pertinent to do more research on studies similar to yours and improve the framing of the topic and discussion.

Response: Thank you very much for your review. We have tried to search recent studies of participatory research methods on CCV health. but there are less studies. Thank you very much for your suggestion.

Thank you very much for your review.